# Patient-Derived Models of Cancer in the NCI PDMC Consortium: Selection, Pitfalls, and Practical Recommendations

**DOI:** 10.3390/cancers16030565

**Published:** 2024-01-29

**Authors:** Amber N. Habowski, Deepthi P. Budagavi, Sandra D. Scherer, Arin B. Aurora, Giuseppina Caligiuri, William F. Flynn, Ellen M. Langer, Jonathan R. Brody, Rosalie C. Sears, Giorgia Foggetti, Anna Arnal Estape, Don X. Nguyen, Katerina A. Politi, Xiling Shen, David S. Hsu, Donna M. Peehl, John Kurhanewicz, Renuka Sriram, Milagros Suarez, Sophie Xiao, Yuchen Du, Xiao-Nan Li, Nora M. Navone, Estefania Labanca, Christopher D. Willey

**Affiliations:** 1Cold Spring Harbor Laboratory, Long Island, NY 11724, USA; habowsk@cshl.edu (A.N.H.); poornim@cshl.edu (D.P.B.); caligiu@cshl.edu (G.C.); 2Department of Oncologic Sciences, Huntsman Cancer Institute, University of Utah, Salt Lake City, UT 84112, USA; sandra.scherer@hci.utah.edu; 3Children’s Research Institute and Department of Pediatrics, University of Texas Southwestern, Dallas, TX 75235, USA; arin.aurora@utsouthwestern.edu; 4Jackson Laboratory, Farmington, CT 06032, USA; bill.flynn@jax.org; 5Division of Oncological Sciences, Oregon Health & Science University, Portland, OR 97239, USA; langere@ohsu.edu; 6Department of Surgery, Oregon Health & Science University, Portland, OR 97239, USA; brodyj@ohsu.edu; 7Department of Molecular and Medical Genetics, Oregon Health & Science University, Portland, OR 97239, USA; searsr@ohsu.edu; 8Ospedale San Raffaele, 20054 Milano, Italy; foggetti.giorgia@hsr.it; 9Department of Internal Medicine, Yale University, New Haven, CT 06520, USA; anna.arnal@yale.edu; 10Department of Pathology, Yale University, New Haven, CT 06520, USA; don.nguyen@yale.edu (D.X.N.); katerina.politi@yale.edu (K.A.P.); 11Terasaki Institute for Biomedical Innovation, Los Angeles, CA 90024, USA; xiling.shen@terasaki.org; 12Department of Medicine, Duke University, Durham, NC 27710, USA; shiaowen.hsu@duke.edu; 13Department of Radiology and Biomedical Imaging, University of California, San Francisco, CA 94158, USA; donna.peehl@ucsf.edu (D.M.P.); john.kurhanewicz@ucsf.edu (J.K.); renuka.sriram@ucsf.edu (R.S.); 14Department of Pediatrics, Lurie Children’s Hospital of Chicago Northwestern University, Chicago, IL 60611, USA; msuarezpalacios@luriechildrens.org (M.S.); sophia.xiao@northwestern.edu (S.X.); yuchdu@luriechildrens.org (Y.D.); xli@luriechildrens.org (X.-N.L.); 15Department of Genitourinary Medical Oncology, David H. Koch Center for Applied Research of Genitourinary Cancers, The University of Texas MD Anderson Cancer Center, Houston, TX 77030, USA; nnavone@mdanderson.org (N.M.N.);; 16Department of Radiation Oncology, University of Alabama at Birmingham, Birmingham, AL 35233, USA

**Keywords:** patient derived models of cancer, organoids, mouse models, sequencing, tumor microenvironment, tumor cells, metastasis

## Abstract

**Simple Summary:**

Over the last two decades, there has been a major shift in cancer research models. What was once dominated by genetically engineered mice models and fast-growing immortalized cancer cell lines has given way to a number of patient-derived models of cancer (PDMCs) that may be more representative of human disease. However, the complexities of these patient-derived models, both in vitro and in vivo systems, influence both data interpretation and interoperability. To address these challenges, the PDMC Consortium within the National Cancer Institute undertook this review describing the past, present, and future use of PDMCs while providing example vignettes with practical recommendations from the collective experience of the consortium members.

**Abstract:**

For over a century, early researchers sought to study biological organisms in a laboratory setting, leading to the generation of both in vitro and in vivo model systems. Patient-derived models of cancer (PDMCs) have more recently come to the forefront of preclinical cancer models and are even finding their way into clinical practice as part of functional precision medicine programs. The PDMC Consortium, supported by the Division of Cancer Biology in the National Cancer Institute of the National Institutes of Health, seeks to understand the biological principles that govern the various PDMC behaviors, particularly in response to perturbagens, such as cancer therapeutics. Based on collective experience from the consortium groups, we provide insight regarding PDMCs established both in vitro and in vivo, with a focus on practical matters related to developing and maintaining key cancer models through a series of vignettes. Although every model has the potential to offer valuable insights, the choice of the right model should be guided by the research question. However, recognizing the inherent constraints in each model is crucial. Our objective here is to delineate the strengths and limitations of each model as established by individual vignettes. Further advances in PDMCs and the development of novel model systems will enable us to better understand human biology and improve the study of human pathology in the lab.

## 1. Introduction

Patient-derived models of cancer (PDMCs) have come to the forefront of preclinical cancer models and are even finding their way into clinical practice as part of functional precision medicine programs. Our recent advances build upon a century of pioneering work in embryology, where early researchers’ curiosity led to the establishment of in vitro and in vivo mammalian model systems. This foundation enabled the development of more advanced models, including those derived from patient diseases. In the 19th century, early research laid the foundation for modern in vitro cell culture. This included Wilhelm Roux’s work maintaining chick embryo cells in a saline buffer and Leo Leob’s experiments isolating guinea pig embryo cells in agar and transplanting them into adult animals. However, the term ‘tissue culture’ first appeared in Ross Granville Harrison’s 1907 publication on culturing nerve cells [1]. In 1910, Burrows and Carrel advanced cell culture by cultivating various embryonic and adult tissues, refining methods and culture media [2]. They created the first “immortal” cell line from a chick embryo heart, which survived from 1912 to 1946. Additionally, in 1910, they were the first to report culturing cancerous tissue [3]. The first modern cell line, “L-cells”, was established in the 1940s by Wilton R. Earle, who used carcinogens to successfully immortalize mouse fibroblast cells [4]. In 1951, cancer research and patient consent ethics were forever changed when George O. Gey and his coworkers created HeLa cells from the tissue of a patient with uterine cervical cancer [5]. HeLa cells fueled a biomedical research revolution for decades and established the field of 2D cell culture human cancer models.

Although spheroids (2D cell lines grown on/in a matrix) had been cultured as far back as the 1950s [6], Mina Bissel’s work in the 1980s highlighted the importance of the extracellular matrix in determining cell fate and behavior and striking biological differences between 2D and 3D cell culture models [7]. In 2009, Hans Clevers’ group published their first paper describing the methodology of culturing murine intestinal stem cells, which self-organized into 3D structures known as “organoids” [8]. Just a few years later in 2011, Clevers’ group published an adapted methodology showing the culture of patient-derived human colon cancer organoids (PDOs) [9]. This led to further research establishing culture methodology enabling the growth of many different human cancers and has revolutionized in vitro patient-derived model systems in the last decade, including applications for personalized medicine [10].

Initial attempts to generate patient-derived xenograft (PDX) models were fraught with difficulties due to immunological rejection, requiring host animals to be immuno-suppressed via steroid treatment at birth or irradiation. Cancer xenotransplantation was reported by J.B. Murphy in 1913, when the Jensen rat sarcoma was grown in chick embryos [11], while mouse-to-rat transplantation was first reported in the 1920s [12]. Transplantation into the anterior chamber of the eyes of rabbits and guinea pigs had higher take rates, likely related to the more immune-privileged site. The first successful human xenotransplantation dates back to at least 1938, when a human breast cancer was transplanted into rabbits. Issues impacting engraftment related to summer temperatures and time from biopsy to transplantation were recognized. However, it was not until 1969 that the heterotransplantation of a human colon carcinoma was subcutaneously (e.g., heterotopic) transplanted by Rygaard and Povlsen into mutant mice with recessive thymic aplasia, the so-called “nude” mouse, which obviated the need to perform neonatal thymectomies [13]. Orthotopic (i.e., into the organ site from which the tumor arose) human tumor xenotransplantation was not reported until the 1980s, and it displayed more clinically relevant features, most notably, metastases [14]. PDX behavior appeared to depend on several factors, including the site of implantation and whether cell suspension or tumor pieces were injected. Although transgenic mice overshadowed PDXs for a considerable time, PDX models have experienced a revival in the past twenty years. Significant repositories of large patient-derived xenografts and organoids have emerged, accompanied by substantial funding initiatives specifically focused on these patient-derived models.

The Patient-Derived Models of Cancer (PDMC) Consortium (members outlined in Table 1), supported by the Division of Cancer Biology in the National Cancer Institute of the National Institutes of Health, seeks to understand the biological principles that govern the various PDMC behaviors, particularly in response to perturbagens, such as cancer therapeutics. Herein, we, representatives from the member institutions of the PDMC Consortium, provide some collective insights regarding both in vitro and in vivo PDMCs, with a focus on practical matters related to developing and maintaining key cancer models. These insights are collected as a series of short vignettes and highlight model systems and the expertise of the PDMC Consortium members. We wish to emphasize that our white paper manuscript is not intended as an exhaustive review of model systems, and it does not provide an in-depth comparison of models across all cancer types. Rather, our primary aim is to showcase the expertise of the members of the NIH PDMC Consortium and to share the valuable insights gained from the model systems that we have worked on. With ongoing advancements in both in vitro and in vivo systems, selecting the right model hinges on the research question and in considering the strengths and limitations of the models, enabling us to better understand human biology and human disease in the lab.

## 2. Model System Options

All member institutions of the PDMC Consortium utilize at least two different model systems, both in vitro and in vivo, which demonstrate advantages and disadvantages depending on the line of investigation. In this section, we provide several examples from among the consortium to highlight various options and key aspects to success in the laboratory.

### 2.1. In Vitro Model Systems

#### 2.1.1. Vignette #1: Development of Location- and Growth-Factor-Specific Pediatric Brain PDXO Models

The study of pediatric brain tumors is challenging in part due to the lack of clinically relevant and molecularly accurate model systems. While most biological and preclinical studies on brain tumors were and continue to be initiated in in vitro model systems before being validated in vivo in animal models, the use of regionally paired models derived from a common patient would greatly facilitate the direct transition of in vitro findings to in vivo discovery and maximize the chances of future clinical success. PDOs are one of the best in vitro model systems [15,16,17,18,19,20], as they preserve a tumor’s cancer stem cell (CSC) pool and maintain a 3D structure with multiple cellular components similar to those found in the original patient tumors [16,21,22,23,24,25]. Limited patient tumor cell supply and low success rates have hindered the development of pediatric brain tumor PDO models. An alternative, called patient-derived xenograft organoids (PDXOs), involves establishing a xenograft model using patient tissue and then generating PDOs, effectively replicating key in vivo tumor biology. We [26,27,28,29,30,31,32,33,34,35,36,37,38,39,40,41] and others [42] have shown that the direct implantation of patient brain tumor cells into matching locations of severe combined immunodeficient (SCID) mice closely replicated the histopathology, invasive growth, CSC pools, and key molecular genetic abnormalities of the original patient tumors.

Since pediatric brain tumor biology is known to be determined/related to tumor location, we hypothesized that the growth of PDOs is dependent on location-related niche factors. A detailed literature search for recent reports on brain tumor PDO development revealed significant differences in the media and growth factor combinations based on location [43,44,45,46,47,48,49,50,51,52]. Brain PDO formation from pluripotent stem cells has suggested a strong dependence of growth factor combinations unique/selective to different brain regions. In PDXO models, organoid formation was confirmed in >75% of the tumors tested, including in 15/19 cerebral tumors, 3/4 cerebellar tumors, and 3/3 brain stem tumors. The incubation of patient tumors resulted in organoid formation from four out of eight cerebral tumors, two out of two cerebellar tumors, and two out of three brain stem tumors. While additional tumors need to be included to validate the overall PDO formation rate, our initial data are very encouraging. They provide strong support for the use of brain-region-matched/selected growth factor combinations for the development of novel PDO models.

#### 2.1.2. Practical Recommendations

Our tumor-site-related growth factor combinations should facilitate the development of PDO models of brain tumors. This work may also be applicable to other solid tumors, where proximity to the vasculature and different regions of organs or neighboring organs could influence the growth factor environment. It is important to consider the epi-tumor spatial information and how this may influence model system generation success and the recapitulation of human biology.

#### 2.1.3. Vignette #2: Preserving the TME Using PDO Variant Technology Micro-Organospheres

One limitation of PDOs is the lack of a tumor microenvironment (TME). In our recent work, we addressed this limitation by developing a new platform for the high-throughput synthesis of “colloidal cancer cells”, termed micro-organospheres (MOSs). MOSs are efficiently formed and make use of limited patient-derived samples (less than 100,000 cells are needed for high-throughput screening) and are uniform in size and content. They can be formed from defined numbers of founder cells per droplet to produce clonal or oligoclonal populations and are in an environment conducive to cellular growth (Matrigel with a high cellular encapsulation density) (Figure 1A). Importantly, MOSs contain and maintain the TME within the individual clones that can be used to perform high-throughput drug screening and tumor profiling [16].

The whole-exome sequencing of patient tumors and derived MOSs showed a similar pattern of amplifications and losses in a copy number variation (CNV) analysis [16]. In addition, driver mutations were also largely consistent between tissue specimens and MOSs among commonly affected genes in colon, breast, lung, and ovarian cancers [16]. Furthermore, MOSs successfully recapitulates tumor heterogeneity and microenvironmental features, including immune cell populations. To compare the tumor and stromal cell types between tissue and derived MOSs, we performed single-cell RNA sequencing (scRNA-seq) between matched lung, colon, and ovarian tumor specimens and derived MOSs. Cells from the matched samples were clustered using Uniform Manifold Approximation and Projection (UMAP) reductions into four groups marked as tumor cells, cancer-associated fibroblasts, lymphoid cells, or myeloid cells, which were concordant between tissue and MOSs (Figure 1B). Flow cytometry and immunofluorescence staining confirmed all major immune cell populations, as well as various T-cell markers in MOSs (Figure 1C). Furthermore, TCR-seq showed that MOSs retained the relative abundance of different TIL (TCR) clones (because the TILs are exposed to tumor cells and their antigens) (Figure 1D), making MOSs a clinically relevant model that is capable of maintaining the TME.

#### 2.1.4. Practical Recommendations

MOS technology stands as a robust contender within in vitro model systems, offering a promising avenue for studying the TME. It serves as both an alternative and a preliminary step preceding in vivo systems, effectively addressing intricate biological inquiries regarding the TME. Although not all models can recapitulate the TME, for some downstream functional assays and specific research questions, it is essential for a complex TME to be studied and, thus, crucial for TME-compatible in vitro models to be developed. MOSs and a few other in vitro options are discussed below, but alternatives are to use an in vivo system or work to advance the field further and develop additional TME models that fit the experimental needs and research question.

#### 2.1.5. Vignette #3: Recreating Pancreas Tissue Structure and TME Using 3D Bioprinting

To better model cancer cells in the context of a complex ecosystem, extrusion-based 3D bioprinting can be used to generate scaffold-free human cancer tissues that exhibit a tissue-like cellular density [53,54,55]. Patient-derived cells can be expanded prior to use in bioprinted models either in culture as a patient-derived cell line (PDCL) or PDOs, or in vivo as a PDX. The patient-derived cancer cells can then be bioprinted such that they are surrounded by commercially available or laboratory prepared primary human stromal cells, including fibroblasts, endothelial cells, and immune cells. These tissues are compatible with the use of patient-derived cells, including autologous stromal cells, allowing for a robust and reproducible model to assess tissue phenotypes and therapeutic response in the context of cancer cell-intrinsic and -extrinsic heterogeneity.

The generation of bioprinted human tissue models of pancreatic cancer serves to fill a gap in the current approaches to patient-derived modeling, allowing for the construction of a rapid (~1 min per printed tissue), manipulable, and reproducible experimental platform that comprises patient-derived cancer cells, along with many components of the microenvironment. For instance, our recent findings suggest that easily manipulated ingredients of the TME, such as glucose [56], oxygen levels [57], and magnesium concentrations [58], can all affect drug efficacy. All these elements can be evaluated using 3D bioprinting. Future work aims to overcome the challenges in incorporating all autologous patient-derived stromal and immune cells, involving ex vivo expansion and using banked peripheral leukocytes. Ongoing efforts include developing reporters for cellular responses, enabling high-throughput drug screening in 3D. The ultimate goal is to validate patient-derived 3D printing assays, such as drug screens, alongside matched patient outcomes from clinical trials. These studies will assess the real value of 3D bioprinting models in translational and clinical research, particularly for tumor systems like pancreatic cancer with a functional TME.

#### 2.1.6. Practical Recommendations

To print tissues, cell mixtures are resuspended in a gelatin–alginate hydrogel to create bioinks that can be extruded in layers to create the desired architecture. Methodologically, the tissues are crosslinked to allow the cells to lay down an extracellular matrix, and upon the reversal of the crosslinks, generally 2 days post-print, the hydrogel is removed, leaving behind a scaffold-free tissue [53]. This enables an endogenous human ECM that is generated by the cells in the printed tissues rather than Matrigel, as used in PDOs. Bioprinted tissues can be maintained in standard tissue culture conditions for over three weeks, during which time the cells self-organize. This is evident in epithelial cells, as well as in the vascular networks that form throughout the tissue. Of note, in our experience, vascular cells are required for tissue health, allowing for larger tissues to survive in culture without necrotic cores. Bioprinting is adaptable, giving the user control over cellular composition, as well as physical, chemical, and spatial attributes, and a downstream analysis can include multiplexed staining approaches or a single-cell analysis of dissociated tissues. The flexibility in printing and analyzing these tissues allows the user to test the response of distinct patient-derived cancer cells to therapies, as well as to assess how changes to the local environment (e.g., hypoxia or the addition of a specific microenvironmental cell type) [57,59] might affect each patient’s therapeutic response.

#### 2.1.7. Vignette #4: Direct Culture of Intact Human Tissue Using Explant Techniques

Approaches to maintaining intact tissues in a physiologic state in vitro have become model systems of interest, and the current work was recently reviewed [60]. This is an advancing field, and there are currently no standard methods for explant cultures. Methods to prepare tissues for explant culture have included simply placing tissue biopsies in culture without processing [61,62,63], teasing apart [64] or manually mincing [65] tissues, and culturing intact tissue slices cut with a vibratome [66] or precision-sliced with specialized microtomes such as a Krumdieck [67,68]. After the tissues are processed using these various means, a variety of substrates are often used to support the tissues ex vivo. The tissues may be simply placed on cell culture plates, embedded in or on an extracellular matrix [69,70], or maintained on sponges or Gelfoam [71,72]. Bioreactors and microfluidic chambers have also been developed to perfuse tissue cultures [73,74].

#### 2.1.8. Practical Recommendations

As for all in vitro models, the medium is a critical factor for the optimal maintenance of the structure and function of tissue cultures. Each component of tissue culture, from the processing to the substrate and the medium, must be empirically optimized for different types of tissue. We developed protocols for the tissue culture of prostate cancer [75] and renal cell carcinoma (RCC) [76]. For both types of cancer, we obtained 8 mm tissue cores from fresh surgical specimens, from which we prepared thin (300 μm) precision-cut slices with a Krumdieck microtome. These tissue slices were then placed on titanium grids in multi-well dishes placed on a unique rotary apparatus that takes the tissue in and out of the liquid–gas phases for the optimal exchange of nutrients and gases. Using a highly enriched medium called Complete PFMR-4A, we successfully maintained the structure and function of the prostate cancer and RCC tissues for 2–7 days. However, we encountered problems when we attempted to use a similar methodology for the tissue slice culture of RCC from PDXs rather than human surgical specimens. It appeared that PDX tumors were sometimes fattier or more cystic than specimens from patients, features that significantly impacted our ability to prepare thin precision-cut slices with the Krumdieck microtome. We found success by manually mincing RCC PDXs and culturing the tissues on gel foam in dishes versus on titanium grids in the rotating apparatus. A histological analysis and an investigation of the proliferation and RCC biomarkers of this PDX revealed that they were well maintained after 45 h of culture. Our pilot study suggests that manually mincing tissues and culturing on gel foam is an alternative approach for culturing tissues that cannot be sliced with a microtome and cultured on titanium grids in a rotating apparatus.

Our experience indicates that optimized conditions developed for the tissue culture of a given type of cancer derived directly from a patient specimen may not suffice for the same cancer tissue derived from a PDX. While tissue slice cultures are widely used in cancer research as one of the most realistic in vitro models [77], precision slicing and culture on grids in a rotating apparatus may not be feasible in some instances. Our study suggests that the explant culture of manually minced tissues on gel foam may provide an alternative approach. As the practice of using ex vivo cultures of tissues derived from PDXs expands [78,79,80,81], others may confront challenges similar to those that we encountered. Our results suggest that relatively simple modifications of the culture conditions may permit the successful cultivation of PDX tissues.

### 2.2. In Vivo Model Systems

#### 2.2.1. Vignette #5: Identification of Circulating Cancer Cells in Patient-Derived Models of Melanoma

PDXs are integral for studying the function of human cancers in vivo, including tumor-forming potential, metastatic potential and patterns, and tumor heterogeneity [82,83,84]. Complex biological processes like metastasis can only be recapitulated in vivo. We developed a xenograft model in which we transplanted melanomas from patients into NOD.CB17-*Prkdc^scid^ Il2rg^tm1Wjl^*/SzJ (NSG) mice [85], and these cells spontaneously metastasized in a manner that correlated with metastatic behavior in patients [86]. Specifically, we observed that melanomas that form distant metastases in patients are likely to spontaneously form distant metastases in NSG mice (efficient metastasizers), while melanomas that are cured by surgery in patients are unlikely to form distant metastases in mice (inefficient metastasizers).

We also observed heterogeneity in the ability of different patient tumors to which they give rise to circulating melanoma cells in the blood of xenografted mice. Efficient metastasizers often give rise to circulating melanoma cells, while inefficient metastasizers do not [86]. This suggests that the ability of melanoma cells to form distant metastases after xenografting into NSG mice is limited by the ability to migrate into, or survive within, the blood. In support of this, melanoma cells experience high levels of oxidative stress when metastasizing through the blood but not when growing as primary subcutaneous tumors [87]. This leads to the death of most metastasizing melanoma cells by ferroptosis, a form of cell death marked by lipid oxidation [88]. The rare cells that survive metastasis upregulate antioxidant pathways, including the folate pathway [87], the pentose phosphate pathway [89], and lactate uptake [90], each of which confer oxidative stress resistance.

Melanoma cells often metastasize to regional lymph nodes before metastasizing through the blood to distant sites [91,92,93]. This appears to be because melanoma cells are protected from oxidative stress in the lymph. While migrating through or growing within lymphatics, they incorporate oleic acid into membrane phospholipids, rendering them ferroptosis-resistant [88]. Thus, melanoma cells that migrate first through lymphatics are more likely to survive subsequent metastasis through the blood because they are ferroptosis-resistant.

#### 2.2.2. Practical Recommendations

We perform xenograft assays by expressing DsRed and luciferase within patient-derived melanomas, and then we inject the cells subcutaneously in NSG mice. Only around 10 cells are required to form subcutaneous tumors in these mice, but an injection of larger numbers of cells accelerates the formation of tumors. While accelerated tumor formation is convenient, it can undermine the ability of the cells to form metastatic tumors because there is less time available for the cells to spontaneously metastasize and grow at distant sites before the mice must be euthanized due to subcutaneous tumor burden. Consequently, we tend to initiate experiments with injections of 100 melanoma cells. Then, we allow the cells to spontaneously metastasize until the subcutaneous tumor reaches 2.0 cm in diameter. Using flow cytometry, we quantify the frequency of melanoma cells in the blood and metastatic disease burden within individual organs via bioluminescence imaging [86]. Melanoma cells are identified as being positive for human HLA and DsRed, and negative for mouse endothelial and hematopoietic markers. We can also quantify the local dissemination of melanoma cells by identifying tumor-draining blood and lymphatic vessels. We inject Evan’s blue dye into the edge of subcutaneously growing melanomas and then allow it to diffuse into a tumor-draining lymphatic or a tumor-draining blood vessel. Then, we insert a 27-gauge Hamilton syringe into the lumen of the blood vessel or the lymphatic, withdraw a few microliters of fluid, pool it among mice, and analyze the number of DsRed^+^HLA^+^ melanoma cells per microliter. Samples must often be pooled from 4 to 10 mice to gain enough cells for analysis. Red blood cells are eliminated from samples using Ficoll sedimentation.

We can also directly test the potential of melanoma cells to survive and form metastatic tumors after an intravenous or intralymphatic injection. We have found that metabolic heterogeneity among melanoma cells influences the ability of the cells to survive upon an intravenous injection: MCT1^high^ cells have an increased ability to withstand oxidative stress and to form metastatic tumors after an intravenous injection [90]. Due to the increased levels of oxidative stress in the blood, larger numbers of cells must be injected into the blood, as compared to subcutaneous injections, in order to induce the formation of tumors (e.g., 400 cells). Apparently, due to the lower oxidative stress in lymphatics, lower numbers of cells can be injected and still form metastatic tumors (50–100 cells) [88].

#### 2.2.3. Vignette #6: Challenges in the Generation of Xenograft Models of Pancreatic Cancer and Selection of the Right Model

Despite great advancements in the generation of in vitro models that can recapitulate specific aspects of tumor biology (discussed elsewhere in this publication), arguably the most informative models from a clinical point of view are the ones in which the effects of treatments or genetic perturbations can be observed systemically. Due to a fast tumor development and straightforward implantation technique, subcutaneous patient xenografts (PDXs) are widely used to model a variety of solid malignancies. However, in recent years, a deeper understanding of the impact of the TME on cancer development and response to therapies has emphasized the need to generate orthotopic models that would more faithfully epitomize the human disease in its entirety. Pancreatic ductal adenocarcinoma (PDAC) is one of the deadliest cancer types for which standard therapies have failed to significantly improve patients’ outcomes [94,95]. This could be partly due to the complex mutational background of PDAC patients, which translates into a wide range of molecular alterations that make the targeting of specific vulnerabilities particularly challenging [96,97]. For this reason, PDAC is one of the most prominent examples of cancers that would benefit from a personalized medicine approach.

To achieve the considerable accuracy level of drug response prediction needed to inform a patient’s therapeutic regime, a patient-derived sample can be engrafted orthotopically into the pancreas to recreate a tumor that displays the same genetic and molecular features of the patient. Currently, two pancreatic transplantation models have achieved a high level of similarity with the original tumor, i.e., orthotopic grafted organoids (OGOs) and intraductal grafted organoids (IGOs) [98,99]. In OGOs, PDOs are transplanted directly into the pancreatic parenchyma, while in IGOs, PDOs are injected in the pancreatic duct and will eventually invade the surrounding tissue. From extensive sequencing efforts on both OGOs and IGOs, as well as more classical transplantation models, such as subcutaneous PDXs, we are now appreciating that orthotopic transplants display a higher degree of transcriptional similarity with the tumor from which the PDO line was derived ([99] and unpublished data). Among all the models considered, IGOs give rise to tumors that not only show the same molecular characteristics of the patient but also display the same histological features. The downside of this is that the tumor follows very similar kinetics to the human disease, and it is therefore slow to develop and difficult to detect at early stages due to the physical location of the pancreas within the abdominal cavity.

#### 2.2.4. Practical Recommendations

For the investigation of PDAC biology, the reliability and accuracy of IGOs in all aspects of tumorigenesis are highly desirable. However, these aspects represent an obstacle for in vivo drug testing that needs to be reckoned with when considering models that show such a high degree of similarity to patients’ tumors. How important it is to pursue a model that is as similar as possible to the human tumor? Is it worth compromising on a faster but less faithful model to obtain clinically relevant information? These and other points revolving around the data turnaround of an in vivo cancer model need to be factored in when we set out to design the best experimental framework. Sometimes, the most accurate model is not necessarily the one that will provide the answers we seek.

Model selection is a complex process, and numerous factors need to be considered. There must be a balance between the faithfulness of the model towards the human biology being investigated and the consideration of cost, time, and necessary expertise. We recommend piloting projects using the simplest and fastest model system and only moving to more advanced and costly models when the previous model systems are unable to answer the biological question or do not recapitulate the necessary biology. Importantly, the more complex the model, generally, the less throughput that can occur, and so the abundance of data that need to be tested/generated also need to be considered. In some cases, the answer of the type of model system is not obvious, but the early identification of the correct model system, or perhaps one of the best in each scenario, will save time and money.

### 2.3. Important Considerations for In Vivo Models: Approaches to Humanized Host Mice for PDXs

Although not a major focus of the PDMC Consortium, several groups are interested in developing humanized murine models, particularly for immuno-oncology studies. There are several approaches to “humanizing” the host for PDXs, but they all essentially utilize a severely immunocompromised mouse that has been transplanted with hematopoietic stem cells (HSCs) following myeloablation. Strategies to improve the model include using HLA-matched allogenic CD34+ or autologous CD34+ HSCs, as well as incorporating human hematopoietic cytokines through transgenic manipulation. Indeed, immune-deficient mice that also harbor [S] the Human Cell Factor (SCF) gene, [GM] Human Granulocyte/Macrophage-Colony Stimulating Factor 2 (GM-CSF), and [3] Human Interleukin-3 (IL-3) transgenes will display increased populations of several human immune cell populations. Even the microbiome can be humanized by using germ-free mouse colonies that undergo human fecal transplantations. NSG and NRG (NOD/RAG1/2^−/−^IL2RG^−/−^) are the most popular choices for mouse strains. One potential advantage of NRG hosts over NSG is that NSG has a non-homologous end-joining repair deficit, while NRG has normal DNA repair, which is an important consideration if using radiation therapy. Humanized models can require extensive time and funds, but for some research questions, they are crucial. Further considerations of humanized mice have been previously published by others [100,101].

## 3. Model System Challenges and Recommendations

While PDMCs provide significant advantages over immortalized model systems, they are not without challenges. In this section, we review some potential issues, particularly related to unintended outgrowths in both in vitro and in vivo systems, and the challenges with multispecies mixtures.

### 3.1. Vignette #7: Detection of Non-Malignant Cells in PDO Cultures

Protocols to establish PDO cultures from solid tumor tissue often require the enzymatic and mechanical digestion of the patient tumor tissue and subsequent embedding in Matrigel. During this first step, there is a possibility that not only tumor cells are embedded into the initial PDO culture but also non-cancerous patient cells. While it might be beneficial to maintain non-cancerous cells for tumor microenvironmental studies [102], for other purposes, the goal is to establish “clean” cancerous PDOs. While many non-cancerous cell types like immune cells will self-eliminate during the first days of in vitro culture or during the first passaging step, in some cases, non-cancerous patient material can form persistent PDOs or cause PDO failure by outcompeting slow-growing malignant cells. Based on our findings, the more common stromal non-cancerous outgrowths display an obvious mesenchymal phenotype, whereas non-cancerous, normal outgrowth cultures display a PDO-like morphology, which is indistinguishable from breast cancer PDOs under the microscope. The outgrowth of non-cancerous PDO cultures has only been observed when the culture was initiated from solid primary breast cancer samples or primary recurrences, never from distant metastases or fluid samples (ascites or pleural effusions). However, regarding PDOs derived from PDACs, normal outgrowth has been observed in PDOs derived from both primary sites and liver metastases.

Preventing normal outgrowth is challenging, and for some tissue types, it is not always possible. For PDOs derived from colorectal cancers, normal outgrowth can be prevented by the absence of Wnt ligands in the culture media, which are not needed for the proliferation of APC-mutant tumors but are necessary for wild-type APC-normal cells [9]. For many other tissue types, there are no clear differences in media conditions that can prevent normal outgrowth. Additionally, due to the often low input material, tumor cell enrichment steps like FACS or magnetic bead sorting are often not viable options to enrich for malignant cells. We have noticed that freshly digested samples derived from solid breast cancer tissue are also especially sensitive to sorting methods, and there are no single cell surface markers that reliably distinguish cancer cells from normal breast cells, so these methods are likely not alternative options to enrich for tumor cells. We found that non-cancerous PDAC PDOs can be passaged for at least five passages or ~84 days in culture. As the use of PDOs in precision medicine advances, it is of critical importance to identify non-cancerous PDOs early so that they are not included in drug response studies.

### 3.2. Practical Recommendations

The early validation of PDOs is a critical basis for the generation of accurate drug screening data. In addition to standard QC methods like short tandem repeat (STR) testing, we recommend screening each PDO line for tumor content prior to performing functional assays and experiments. If enough material is available, traditional DNA extraction from lysates can be performed [103]. For samples with limited PDO material or if a culture needs validation at a very early timepoint, we recommend the isolation of cell-free DNA from conditioned PDO culture media. Using the extracted DNA, a copy number variant analysis can be performed to check for tumor-site-specific cancer-related variants or ddPCR for frequent cancer mutations. In addition, we utilize methyl patch PCR or whole-genome methylation to look at the methylation status of a validated panel of breast-cancer-specific methylation sites developed by the Varley lab, University of Utah (unpublished). Establishing the purity of PDO culture lines, as well as determining the amount of normal outgrowth, is crucial in validating PDO model systems.

### 3.3. Vignette #8: Development of PDX Biobank Derived from Prostate Cancer

In prostate cancer (PCa), progress in understanding the determinants of metastasis and therapy resistance has been hampered by a lack of models representative of the clinical spectrum and biologic complexity of PCa. Recently, PDMCs (PDXs and PDOs) have been developed and have led to therapeutically relevant approaches [104,105,106,107,108,109] The success of these models and recent large-scale genomic studies that have identified deregulated pathways in metastatic castration-resistant PCa further drives the impetus to understand and improve the utility of PDMCs in addressing clinical gaps that limit progress.

To address the lack of clinically annotated relevant models of PCa, particularly models that capture the progression to metastasis and castration resistance over time, we developed PDXs from clinically annotated tumor specimens derived from men with PCa taken at a single time (non-longitudinal) and different times (longitudinal studies) and from different areas of the same tumor during progression. These models have been proven to mimic the donor human PCa [110,111] and to be useful for drug testing [112]. We successfully established 152 PDX lines out of 581 attempts. Of those, we established 85 PDXs using tissue derived from primary PCa out of 323 attempts (~26%); 25 PDXs using tissue derived from bone metastases out of 94 attempts (~26%); and 41 PDXs using tissue from other metastases out of 164 attempts (~25%).

### 3.4. Practical Recommendations

When developing a PDX biobank, it is important to consider the success rates for a given cancer type and how they might change over time. Our success rate has changed over time because we shifted our focus to primary tissue and because of changes to standard of care (second-generation anti-androgens seem to decrease the take rate). Other factors, such as early detection, changes in systemic or radiation therapy, and the timing of surgery relative to disease progression/treatment, could change and potentially influence PDX success rates. Our initial overall success was 30–40%, and it is currently 26%. At this current rate, we need to collect 4× the number of samples compared to models that we hope to generate.

A genomic validation of models is crucial to ensure that PDXs recapitulate the mutational landscape and are clinically relevant. Advanced PCa is characterized by somatic mutations, gene fusions, chromosomal rearrangements, and copy number variations (CNVs) [113,114]. With that in mind, we selected 45 PCa PDX models derived from 39 patients’ tumors that reflect the various morphologic groups, stages, and treatment statuses of the disease. In these models, we performed whole-genome sequencing (WGS), targeted sequencing for 263 gene mutations implicated in the pathogenesis of solid cancers (T200), and RNA sequencing. Importantly, the molecular and morphological analyses of each PDX were performed in representative samples of a single tumor. This design facilitates the integrated analysis of the different approaches of genomic analysis with morphologic and immunoassays results. This type of genomic investigation is an important aspect of building and annotating a PDX biobank. In addition, PDOs can be developed from PDXs. Knowing the genomics of these models would allow us to ask scientific questions by editing the PDXO models, which can be integrated in vitro and/or subsequently integrated into mice to analyze effects in vivo.

### 3.5. Vignette #9: Detection of Mouse Cell Contamination in Human PDX Models

The establishment and maintenance of PDXs require the use of an immunocompromised host. The more severe the immunocompromise, the theoretically higher the take rate of the PDX. For example, NOD *scid* gamma (NSG) mice have a severe combined immune deficiency (scid) due to a mutation in DNA PK (*Prkdc^scid^*), a key protein in non-homologous end-joining DNA repair, which renders NOD/ShiLtJ genetic background mice devoid of mature B and T cells. In addition, these mice have an interleukin 2 receptor gamma chain knockout (*IL2rg^null^*) mutation, which prevents normal cytokine signaling, so they also lack functional natural killer cells. As such, these mice are frequently used to establish PDXs or for humanization approaches due to their high engraftment rate. Unfortunately, there are some potential issues with serial PDX passage in these models. For example, some tumors are highly invasive, particularly when implanted orthotopically. When tumors are macrodissected and dissociated for subsequent passaging, an admixture of mouse cells can contaminate the implanted tissue. With the DNA repair deficit inherent within the mouse host, these contaminating mouse cells can theoretically undergo further mutations, which could support oncogenesis. We have seen spontaneous 100% mouse sarcomas develop from pediatric brain tumor PDXs when maintained in Cb17-SCID mice [115]. This issue has been observed in other solid tumor PDX models, and it is critical to credential and monitor models when using these hosts.

### 3.6. Practical Recommendations

Even when using less severe murine hosts, such as athymic nude mice (our preferred host for GBM PDX), there is always a concern that the PDX may undergo significant changes or become so contaminated with mouse cells that subsequent passages will be difficult. Credentialing or benchmarking the models to confirm human versus mouse proportions should be performed. There are several methods available, including simple qPCR [116], mouse- and human-specific antibodies, next-generation sequencing pipelines [117], and methylation profiling. We routinely freeze down a portion of tumor tissue chunks and dissociated cells with each passage. As with any cell line, PDXs should be periodically evaluated using STR assays to confirm tumor line authenticity.

### 3.7. Vignette #10: Bioinformatic CHALLENGES in Single-Cell Analyses of Mixed-Species Model Systems of Cancer

Characterizing model systems with single-cell sequencing technologies can be challenging, especially in those comprising multiple species, such as xenograft models. Most microfluidic-based dissociative single-cell RNA-seq (scRNA-seq) technologies assume that each droplet contains mRNA from a single cell. In reality, each droplet can contain a mixture of cellular mRNA (from one or more cells) and ambient mRNA. The amount of ambient mRNA and the ambiguity of its source increase as the viability of the source material decreases. The presence of ambient mRNA poses several data processing and analysis challenges, specifically related to appropriately mapping sequencing reads to a reference transcriptome and interpreting the resulting data downstream.

Most data processing strategies of multispecies data involve mapping to a chimeric reference, a concatenation of two species’ reference genomes. In the single-cell field, these have only seen substantial use in technology benchmark experiments to quantify doublet rates in so-called barnyard experiments [118], where human and mouse cell lines are pooled together into a single library (typically to demonstrate the multiplet rate for new technologies). Disparities in the annotation quality or assembly versions in the single-species genomes comprising chimeric reference genomes can lead to mapping challenges, including the complete mis-mapping of certain genes, particularly pseudogenes and lncRNAs. Moreover, scRNA-seq technologies employ short sequencing reads, typically targeting the 5′ or 3′ end, with only one biological read in the mate pair (the other is used to store cellular barcoding information), and this leads to lower transcriptomic mapping rates than the true paired-end sequencing normally employed in bulk RNA-seq or SMART-seq. While most processing pipelines discard sequencing reads that map equally well to the transcriptomes of both species, we have observed that short transcriptomic fragments from one species can be confidently *mis*-mapped to their ortholog in another species; e.g., aligning human cell lines to a chimeric human–mouse reference will yield confidently mapped mouse transcripts. For example, the alignment of human cells from human PDAC tumors and primary PDO cultures to a chimeric human–mouse reference has resulted in the nontrivial mis-mapping of human transcripts to the mouse portion of the chimeric GRCh38-mm10 reference genome provided by 10× Genomics. Many of these hits match human transcripts to orthologous mouse genes that appear to simply have a higher 3′-end similarity to the observed mouse transcript than to the annotated human reference sequences.

The annotation quality of mitochondrial and ribosomal genes differs in the two references comprising the chimeric reference genome, and this may lead to the complete mis-mapping of the mtRNA and rRNA transcripts that are captured in large quantities via platforms like 10× Genomics. For example, the putative human 18s rRNA ortholog of mouse 18s rRNA Gm42418 (mm10; Rn18s-rs5 mm11) is not annotated in any build of GRCh38. In the PDAC PDO models referenced above, this led to hundreds of thousands of unique transcripts of Gm42418 in human cells. Therefore, the presence of multispecies ambient mRNA and mapping inconsistencies with chimeric reference transcriptomes may subtly affect downstream conclusions derived from standard differential expression or marker gene detection analyses; careful experimental planning and analysis strategies should be in place to account for these challenges.

There exist several tools to partially mitigate these issues, specifically Xenome [119]—a tool to separate graft from host reads in bulk RNA-seq—as well as its adaption to scRNA-seq and XenoCell [120]. Both tools use a multispecies set of reference files to partition sequencing reads into reads belonging to the host or graft. In the case of XenoCell, this is then used to quantify the mixing or contamination rate for each cell. Using the output of these tools, one can restrict the analysis to only cells with a minimal amount of ambient contamination. Unfortunately, these tools do not alleviate the challenges with the mis-mapping of ambient mRNA. There exist other software tools that attempt to remove ambient mRNA from scRNA-seq data that can be useful to an extent, but, ultimately, there is very little one can do to completely remove ambient mRNA from each cell.

The challenges posed by ambient mRNA are compounded in comparative analyses between model systems or model systems with the original primary sample. Human cell transcriptomes derived from a xenograft model will be contaminated with mouse ambient mRNA (which is possibly mis-mapped to human orthologs), whereas human cells derived from a purely human microenvironment will contain human ambient contamination. Typical analyses integrate multiple datasets together in order to jointly define cell populations; this can be a critical validation that confirms that cell populations represented in the primary tissue are also represented in the model system. Due to differing ambient mRNA profiles, it may be exceedingly difficult to directly integrate these seemingly disparate transcriptomes and confidently identify a shared cellular identity using the standard scRNA-seq analysis toolkit. Alternative strategies to understand cell-type similarity between populations confounded by differing cell states, ambient mRNA contamination, and other factors—such as gene modules or different matrix decomposition methods—are an active area of research.

### 3.8. Practical Recommendations

Before beginning experiments and comparative analyses of model-derived scRNA-seq, we recommend the following steps:Develop robust sample collection, sample preparation, and library generation plans to maximize viable tissue, maximize post-dissociation cell viability, and minimize the time spent on the bench.Get to know your references: align both mouse-only and human-only data to your chimeric reference to identify commonly mis-mapped transcripts; construct a list of ambiguous genes to exclude from downstream analyses.Align sequencing reads to a chimeric reference first to partition cell barcodes into human, mouse, or ambiguous categories. This can be carried out using a standard aligner and a “barnyard” chimeric reference or using tools such as XenoCell [120].Select barcodes with low mixture rates, and realign reads corresponding to those barcodes to their respective species reference genome.Take caution when attempting to integrate data derived from different microenvironments, as ambient contamination may introduce artifacts and increase false-positive differentially expressed genes or nonsensical marker genes. Think about leveraging alternative dimensionality reduction tools, such as NMF, to integrate observations along shared sets of gene programs.

## 4. Conclusions

The selection of an appropriate patient-derived model begins with an assessment of requirements versus “luxuries” for the scientific question. For example, some features, such as compatibility with high-throughput robotic platforms, may be more of a secondary concern compared to defined extracellular matrix components when examining tumor cell invasion. However, the major decision point is usually deciding between in vitro and in vivo models. Like most features in research, there is often an inverse relationship between cost and complexity, which applies to PDMCs. In vivo models, in general, are more costly and time-consuming (compared to their in vitro counterparts) and require sufficient infrastructure to establish and maintain the models. For example, in humanized mouse models, it can be quite expensive to set up and maintain aspects such as dedicated germ-free gnotobiotic facilities for human gut microbiome transplants. Additionally, the models involve complexities related to collecting, banking, and transplanting CD34+ patient- or HLA-matched hematopoietic stem cells for the humanization of myeloablated immunocompromised mice. So, it is prudent to consider or at least start with the least complex, quickest, and cheapest model that will enable the research question of interest to be investigated. In some cases, initial questions can be answered, or hypotheses eliminated, with 2D cell lines. When needed, follow-up work can be performed in more complex and expensive systems, moving from PDOs to mice, for example. In other scenarios, the research questions might necessitate complex explants, bioprinting, or humanized mouse models. It remains crucial to acknowledge that the mere availability of a model system does not inherently justify its use. The most elegant and convincing science is achieved when the model system fits the research question. Figure 2 provides a summary of options and considerations when selecting a suitable PDMC.

The PDMC research space is rapidly developing in terms of availability and model systems. This manuscript highlights the models and cancer types of the NCI PDMC Consortium members; however, there are many other model systems available and others in development [25,60,121,122,123,124,125,126,127]. There are also numerous challenges (and opportunities) that the cancer research community will face in the coming years. Currently, there is considerable effort to “standardize” certain PDMCs while adopting standards of minimal information to support reproducibility and collaboration. We are beginning to see the fruits of this labor, with repositories such as HCMI-Leidos, NCI PDMR, and PDCM Finder gaining prominence. However, even connecting PDMC development with clinical and translational practices is fraught with regulatory hurdles (e.g., IRB and HIPAA), biosafety, and infrastructure/pipeline needs. We encourage future consortia and similar group research efforts to address these issues in the coming years.

## Figures and Tables

**Figure 1 cancers-16-00565-f001:**
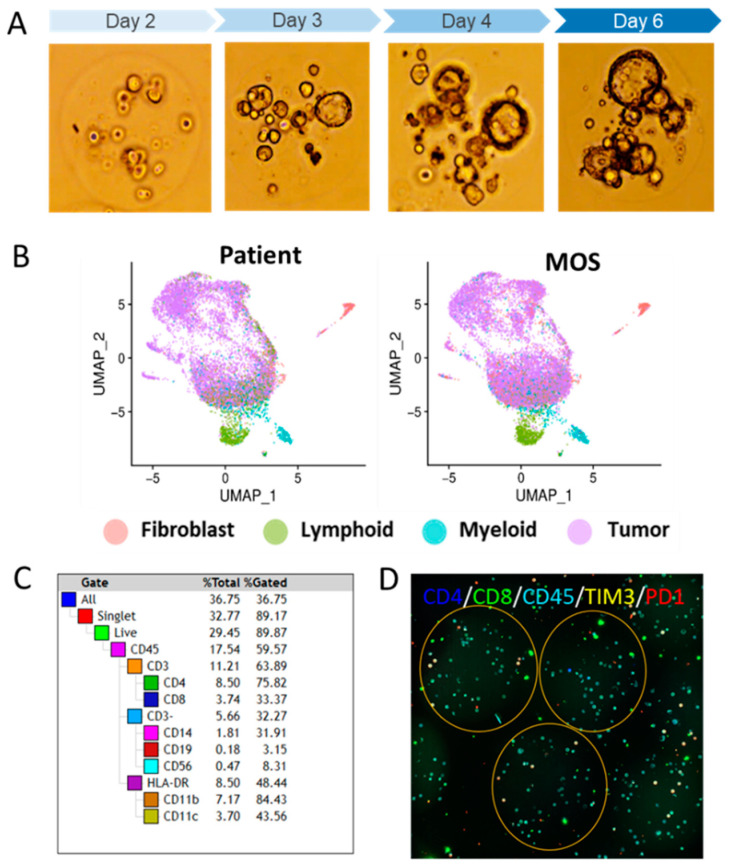
MOSs retain patient TME. (**A**) Tumor spheres form rapidly in MOSs. (**B**) scRNA-seq of MOSs developed from lung cancer tissue retains all major cell clusters, including tumor, fibroblast, lymphoid, and myeloid cells. (**C**) Flow cytometry shows the presence of all major immune cell populations in MOSs. (**D**) Immunofluorescence shows expression of T-cell markers in MOSs.

**Figure 2 cancers-16-00565-f002:**
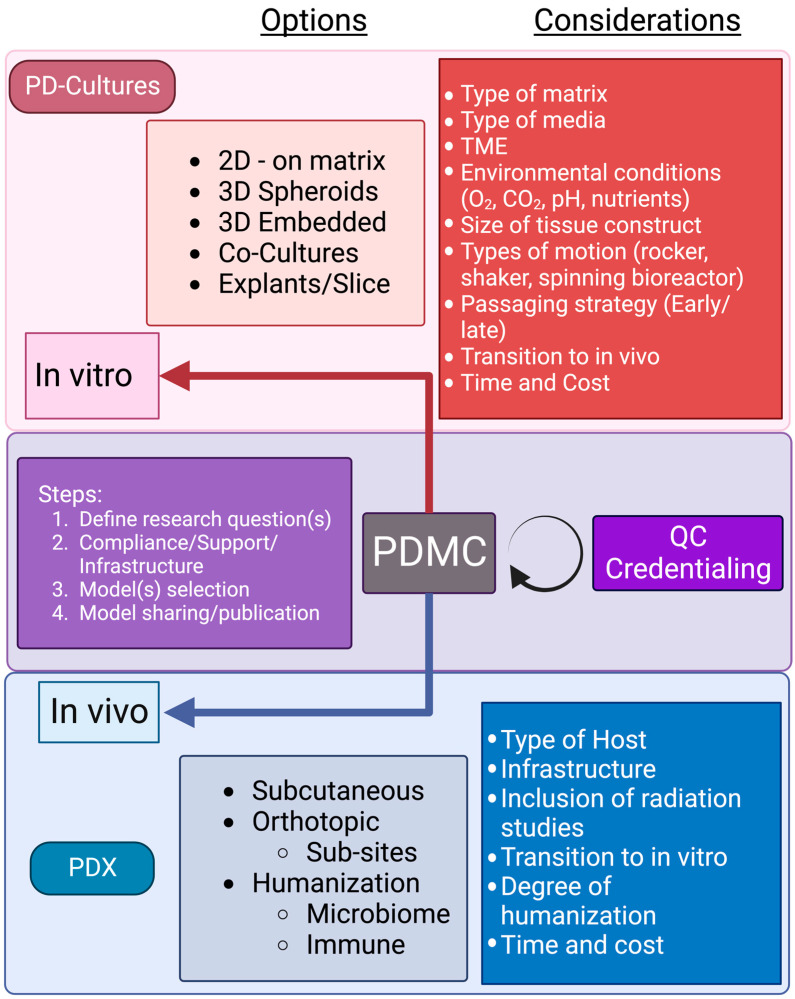
Options and considerations when selecting a PDMC. There is a diverse array of PDMCs that can be used to investigate a defined research question. PDMCs require consideration of research compliance/ethics/institute support/infrastructure that can influence model selection. Model systems are primarily split between in vitro and in vivo systems, and each has its own value and additional methodological considerations. Additional important steps include model validation, namely, quality control and credentialing (benchmarking), and eventual model sharing and publication to further advance the PDMC community and cancer biology research. Created with BioRender.com.

**Table 1 cancers-16-00565-t001:** Overview of PDMC Consortium groups and funded projects.

Institution	Principal Investigator(s)	Project Title
Cold Spring Harbor Laboratory & Jackson Laboratory	David Tuveson and Paul Robson	CSHL-JAX Patient-Derived Models of Pancreatic Cancer as Systems for Investigating Tumor Heterogeneity
Duke University	Xiling Shen and Shiaowen D. Hsu	Epigenomic Reprogramming in Patient Derived Models of Colorectal Cancer
Lurie Children’s Hospital of Chicago	Xiao-Nan Li	Matching Panels of in vivo and in vitro Model System of Pediatric Brain Tumors
Oregon Health & Science University	Rosalie C. Sears, Jonathan Brody, Lisa M. Coussens, and Emek Demir	Comparative Analysis between Patient-derived Models of Pancreatic Ductal Adenocarcinomas and Matched Tumor Specimens
University of Alabama at Birmingham	Christopher D. Willey, Jake Y. Chen, Xiangqin Cui, G. Yancey Gillespie, Anita Hjelmeland, and Raj K. Singh	Biological Comparisons Among Three Derivative Models of Glioma Patient Cancers Under Microenvironmental Stress
University of California, San Francisco	John Kurhanewicz and Donna M. Peehl	Metabolic Imaging Comparisons of Patient-Derived Models of Renal Cell Carcinoma
University of Texas MD Anderson Cancer Center	Nora Navone, Yu Chen, and Phillip A. Futreal	Patient-Derived Models of Prostate Cancer for Personalized Medicine
University of Texas Southwestern Medical Center	Sean J. Morrison	The Metabolic Regulation of Melanoma Metastasis
University of Utah	Bryan E. Welm, Gabor T. Marth, and Katherine E. Varley	Longitudinal Models of Breast Cancer for Studying Mechanisms of Therapy Response and Resistance
Yale University	Katerina A. Politi and Don X. Nguyen	Uncovering the Biology of Resistance to TKI in EGFR Mutant Lung Cancer

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
