# Peer review of "Patient-Derived Models of Cancer in the NCI PDMC Consortium: Selection, Pitfalls, and Practical Recommendations"

_cancers, 2024, doi:10.3390/cancers16030565_

Round 1

Reviewer 1 Report

Comments and Suggestions for Authors

This paper presents a very nice review concerning the past, present and future use of PDMCs with practical recommendations from National Cancer Institute specialists.

The structure of the manuscript is well organised, and the examples presented are well discussed. I must say, I thoroughly enjoyed reading this article.

An important aspect covered in the work is the presentation of the advantages, disadvantages, and challenges associated with different cancer models. Figure 1 provides a clear and concise summary of the entire work. Congratulations !!!!

I strongly support publishing the article with minor linguistic changes. Some paragraphs in the abstract, the introduction section and chapter 2 (Model System Option) are too long and thus incomprehensible.

Comments on the Quality of English Language

I strongly support publishing the article with minor linguistic changes. Some paragraphs in the abstract, the introduction section and chapter 2 (Model System Option) are too long and thus incomprehensible.

Author Response

“This paper presents a very nice review concerning the past, present and future use of PDMCs with practical recommendations from National Cancer Institute specialists. The structure of the manuscript is well organised, and the examples presented are well discussed. I must say, I thoroughly enjoyed reading this article. An important aspect covered in the work is the presentation of the advantages, disadvantages, and challenges associated with different cancer models. Figure 1 provides a clear and concise summary of the entire work. Congratulations !!!!”

We thank Reviewer 1 for their enthusiastic support! We are proud of being part of this NIH PDMC consortium and excited to share our work with a broader audience.

“I strongly support publishing the article with minor linguistic changes. Some paragraphs in the abstract, the introduction section and chapter 2 (Model System Option) are too long and thus incomprehensible. I strongly support publishing the article with minor linguistic changes. Some paragraphs in the abstract, the introduction section and chapter 2 (Model System Option) are too long and thus incomprehensible.”

In response we have done additional editing and refining of the manuscript. We have removed text from the introduction, and chapter 2 and worked to make the language more concise. The edited portions are yellow highlighted in the manuscript.

Reviewer 2 Report

Comments and Suggestions for Authors

Habowski et al. presented a review article on various models of cancer. Authors introduced various in vitro and in vivo model systems, discussed applications and limitations, and made several practical recommendations. Although this review is relevant and interesting to the readers in this field, most of the topics covered are superficial. Authors must address the following comments.

1.     Authors discussed various interesting topics in both in vitro and in vivo model systems, but most of the topics covered are superficial. For example, in case of MOS, authors discussed their data on the lung cancer. The data is compelling, but it lacks thorough review of similar work in other cancers by other groups and whether or not similar observations were made. If there are differences, and why such discrepancies are observed. Similarly, other optics covered in this report also lack in-depth review. I suggest authors to do a thorough review of literature on various model systems discussed in the report, covering a broad array of cancer types.

2.     Authors discussed briefly on humanized mouse PDX model. I find this discussion very limited and superficial. Authors must thoroughly discuss this topic by covering various aspects. This is an emerging area, potentially interesting a wider audience in the field.

Comments on the Quality of English Language

Appropriate.

Author Response

“Habowski et al. presented a review article on various models of cancer. Authors introduced various in vitro and in vivo model systems, discussed applications and limitations, and made several practical recommendations. Although this review is relevant and interesting to the readers in this field, most of the topics covered are superficial. Authors must address the following comments.

  1. Authors discussed various interesting topics in both in vitro and in vivo model systems, but most of the topics covered are superficial. For example, in case of MOS, authors discussed their data on the lung cancer. The data is compelling, but it lacks thorough review of similar work in other cancers by other groups and whether or not similar observations were made. If there are differences, and why such discrepancies are observed. Similarly, other optics covered in this report also lack in-depth review. I suggest authors to do a thorough review of literature on various model systems discussed in the report, covering a broad array of cancer types.”

We thank Reviewer 2 for this feedback, and we acknowledge this is a limitation of this manuscript. This manuscript is not intended to be a thorough review of model systems and does not include in-depth comparison of models across all cancer types. Instead, we highlight the expertise of members of the NIH PDMC consortium and the valuable lessons we have learned from the model systems we have worked with. However, to address the concerns of Reviewer 2 we have included several additional references and text pointing out the caveats of this manuscript. The added text is highlighted in cyan in the manuscript. 

  1. “Authors discussed briefly on humanized mouse PDX model. I find this discussion very limited and superficial. Authors must thoroughly discuss this topic by covering various aspects. This is an emerging area, potentially interesting a wider audience in the field”

We too felt discussion of humanized mouse PDX was important, and thus it was mentioned despite not being a focus of any members of the NIH PDMC consortium. To address this feedback from Reviewer 2 we have added some additional text and references (highlighted in cyan). We are unfortunately not able to expand further (due to page limitations and the fact that other reviewers have asked for us to be more concise), but we hope the additional references will act as a springboard for those interested in learning more.

Reviewer 3 Report

Comments and Suggestions for Authors

Thank you very much for this review article on patient derived models of cancer. I really enjoyed reading the overview of the available methods and the practical recommendations on how to go about them. I am wondering how come there are no established PDMC's for head and neck cancers ( at least for the squamous cell carcinomas)?

Author Response

“Thank you very much for this review article on patient derived models of cancer. I really enjoyed reading the overview of the available methods and the practical recommendations on how to go about them. I am wondering how come there are no established PDMC's for head and neck cancers (at least for the squamous cell carcinomas)?”

We thank Reviewer 3 for taking the time to review our manuscript and providing feedback. In this manuscript we focused on the model systems and expertise of only the members of the NIH PDMC consortium, and currently there are no members focused on head and neck cancers (HNC). As a result, we regret there was no discussion on HNC PDMCs. That being said, there are many groups working on HNC PDMCs, and we are providing a few links to other manuscripts below. Reviewer 2 brought up our limited coverage of broad cancer types and we have included some additional text in the manuscript to acknowledge that we were unable to highlight all cancer types and PDMCs.

Chaves, P., Garrido, M., Oliver, J. et al. Preclinical models in head and neck squamous cell carcinoma. British Journal of Cancer 128, 1819–1827 (2023). https://doi.org/10.1038/s41416-023-02186-1

Basnayake, B.W.M.T.J., Leo, P., Rao, S. et al. Head and neck cancer patient-derived tumouroid cultures: opportunities and challenges. British Journal of Cancer 128, 1807–1818 (2023). https://doi.org/10.1038/s41416-023-02167-4

Schuch, L.F., Silveira, F.M., Wagner, V.P. et al. Head and neck cancer patient-derived xenograft models – A systematic review. Critical Reviews in Oncology/Hematology 155, 103087 (2020).  https://doi.org/10.1016/j.critrevonc.2020.103087